# A generalized class of estimators for sensitive variable in the presence of measurement error and non-response

**Erum Zahid**[1]*, **Javid Shabbir**[2], **Sat Gupta**[3], **Ronald Onyango**[4], **Sadia Saeed**[1]

**1** Department of Applied Mathematics & Statistics, Institute of Space Technology, Islamabad, Pakistan, **2** Department of Statistics, Quaid-i-Azam University, Islamabad, Pakistan, **3** Department of Mathematics & Statistics, University of North Carolina at Greensboro, Greensboro, NC, United States of America, **4** Department of Applied Statistical, Financial Mathematics and Actuarial Science Jaramogi Oginga Odinga University of Science and Technology, Bondo, Kenya

\* erumzahid22@gmail.com

**Data Availability Statement:** All relevant data are within the paper and its Supporting information files.

## Abstract

In this paper, a general class of estimators is proposed for estimating the finite population mean for sensitive variable, in the presence of measurement error and non-response in simple random sampling. Expressions for bias and mean square error up to first order of approximation, are derived. Impact of measurement errors is examined using real data sets, including the survey conducted at Quaid-i-Azam University, Islamabad. Simulated data sets are also used to observe the performance of the proposed estimators in comparison to some other estimators. We obtain the empirical bias and MSE values for the proposed and the competing estimators.

## 1 Introduction

In survey sampling, if the variable of interest is sensitive in nature, the chance to get incorrect information increases. The problem of measurement error is usually ignored during the sensitive surveys and the assumption is made that the information obtained is free from error. Another important factor in surveys is non-response, which may arise due to refusal by respondents to give the information, or for not being at home. Usually measurement error and non-response are studied separately. In reality, when the variable of interest is sensitive, the respondents hesitate to provide the personal information, which gives rise to non-response. Many researchers have studied the problem of non-response, including [1–9]. In survey sampling, when the variable under study contains social stigma, the respondents are not comfortable to provide their personal information. Direct survey on sensitive questions increases the response bias. [10] introduced the randomized response technique (RRT), which reduces the possible response bias by insuring the privacy of the respondents. For estimation of mean of a sensitive quantitative variable, the Randomized Response Model (RRM) was used by [10] and [11]. Further work in this are done by [12–21], among others.

Many researchers have dealt with the problem of measurement error for estimating the population mean. For more details, see [22–27], etc. Recently few researchers have studied the

**Funding:** The authors(s) received no specific funding for this work.

**Competing interests:** No authors have competing interests.

problem of measurement error and non-response together; for example [28–33] have discussed the problem of measurement error and non-response under stratified random sampling.

In many cases, the researchers who have studied measurement error, have ignored the presence of non-response and randomized response particularly when using randomized response. In this study, we have proposed a class of estimators for the population mean of a sensitive variable in the presence of measurement error and non-response simultaneously, under simple random sampling. The efficiency of the suggested class of estimators as compared to the existing estimators is shown using simulated and real data sets.

Let $\Omega = \{\Omega_1, \Omega_2, \ldots, \Omega_N\}$ be a finite population of size $N$. Suppose that a sample of size $n$ is drawn from $\Omega$ by using simple random sampling without replacement. We assume that a population of size $N$ consists of two mutually exclusive groups: $N_1$ (respondents) and $N_2$ (non-respondents). After selecting the sample, we assume that $n_1$ units respond and $n_2$ units do not respond. We select a sub-sample of size $k$, $\left(k = \frac{n_2}{g}, g > 1\right)$ from the $n_2$ non-responding units.

Let $Y$ be the sensitive study variable, which is not observed directly and $X$ be a non-sensitive auxiliary variable which has positive correlation with $Y$. Let $R_x$ be ranks of the auxiliary variable $X$. Let $S$ be a scrambling variable which is independent of $Y$ and $X$. We assume that $S$ has zero mean and variance $S_s^2$. The respondent is asked to give a scrambled response for the study variable $Y$, given by $Z = Y + S$, and is asked to provide a true response for $X$.

Let $(z_i^*, y_i^*, x_i^*, r_{x_i}^*)$ be the observed values and $(Z_i^*, Y_i^*, X_i^*, R_{x_i}^*)$ be the actual values corresponding to the $i^{th}(i = 1, 2, \ldots n)$ sampled unit with $R_{x_i}^*$ being the corresponding ranks of $X_i^*$. Then the measurement errors are: $Q_i^* = z_i^* - Z_i^*$, $V_i^* = x_i^* - X_i^*$ and $T_i^* = r_{x_i}^* - R_{x_i}^*$. Note that $Y$ is not observed directly, so we consider measurement error only on its scrambled version $Z$. Let $S_Z^2$, $S_X^2$ and $S_{R_x}^2$ be the population variances of the variable $Z$, $X$ and $R_x$ respectively. Let $S_{Z(2)}^2$, $S_{X(2)}^2$ and $S_{R_x(2)}^2$ be the population variances of the variable $Z$, $X$ and $R_x$ respectively for the non-responding units. Let $S_Q^2$, $S_V^2$ and $S_T^2$ be the population variances associated with measurement error in the variables $Z$, $X$ and $R_x$ respectively. Let $S_{Q(2)}^2$, $S_{V(2)}^2$ and $S_{T(2)}^2$ be the population variances associated with measurement error in the variables $Z$, $X$ and $R_x$ respectively for the non-responding units. Let $\rho_{ZX}, \rho_{ZR_x}, \rho_{XR_x}$ be the population coefficients of correlation for respondents and $\rho_{ZX(2)}, \rho_{ZRx}(2), \rho_{XRx}(2)$ be the population coefficients of correlation for non-responding units.

The layout of paper is as follows: In Section 2, some existing estimators of the finite population mean are given. In Section 3, a generalized class of estimators is suggested for the finite population mean by incorporating both measurement error and non-response information simultaneously. Efficiency comparison is also presented. Numerical results and a simulation study are presented in Section 4. Some concluding remarks are given in Section 5.

## 2 Some existing estimators in literature

In this section, we consider the following existing estimators.

### 2.1 Hansen and Hurwitz (1946) estimator

In simple random sampling, Hansen and Hurwitz (1946) estimator for population mean $\bar{Y}$, is given by

$$\bar{y}_{HH}^{*'} = \bar{z}^*, \qquad (1)$$

where $\bar{z}^* = \left(\frac{n_1}{n}\right)\bar{z}_{n_1} + \left(\frac{n_2}{n}\right)\bar{z}_k$.

Here $\bar{z}_{n_1} = \frac{1}{n_1}\sum_{i=1}^{n_1} z_i$ and $\bar{z}_k = \frac{1}{k}\sum_{i=1}^{k} z_i$ are the sample means based on $n_1$ of responding units and $k$ of the $n_2$ non-responding units, respectively.

The variance of $\bar{y}_{HH}^{*'}$, is given by

$$Var(\bar{y}_{HH}^{*'}) = A^{*'}, \tag{2}$$

where $A^{*'} = \lambda_2(S_Z^2 + S_U^2) + \theta(S_{Z(2)}^2 + S_{U(2)}^2)$, $\theta = \frac{P_2(g-1)}{n}$, $P_2 = \frac{N_2}{N}$, $\lambda_2 = \left(\frac{1}{n} - \frac{1}{N}\right)$.

## 2.2 Ratio estimator

The usual ratio estimator under simple random sampling, is given by

$$\bar{y}_R^{*'} = \frac{\bar{z}^*}{\bar{x}^*}\bar{X}, \tag{3}$$

where $\bar{X} = \frac{1}{N}\sum_{i=1}^{N} x_i$ is the known population mean and $\bar{x}^{*'} = \bar{X} + \frac{1}{n}(\delta_X^* + \delta_V^*)$ is the sample mean (see Eq 22).

The bias and mean square error of $\bar{y}_R^{*'}$, are given by

$$B(\bar{y}_R^{*'}) \cong \frac{1}{\bar{X}}[R'B^{*'} - C^{*'}] \tag{4}$$

and

$$MSE(\bar{y}_R^{*'}) \cong [A^{*'} + R'^2 B^{*'} - 2R'C^{*'}], \tag{5}$$

where
$R' = \frac{\bar{Z}}{\bar{X}}$,
$B^{*'} = \lambda_2(S_X^2 + S_V^2) + \theta(S_{X(2)}^2 + S_{V(2)}^2)$,
$C^{*'} = \lambda_2\rho_{ZX}S_Z S_X + \theta\rho_{ZX(2)}S_{Z(2)}S_{X(2)}$.

## 2.3 Product estimator

The usual product estimator under simple random sampling, is given by

$$\bar{y}_{Pr}^{*'} = \bar{z}^* \frac{\bar{x}^*}{\bar{X}}. \tag{6}$$

The bias and mean square error of $\bar{y}_{Pr}^*$, are given by

$$B(\bar{y}_{Pr}^{*'}) \cong \frac{C^{*'}}{\bar{X}} \tag{7}$$

and

$$MSE(\bar{y}_{Pr}^{*'}) \cong [A^{*'} + R'^2 B^{*'} + 2R'C^{*'}]. \tag{8}$$

## 2.4 Bahl and Tuteja (1991) estimator

Bahl and Tuteja (1991) estimator under simple random sampling, is given by

$$\bar{y}_{BT}^{*'} = \bar{z}^* \exp\left(\frac{\bar{X} - \bar{x}^*}{\bar{X} + \bar{x}^*}\right). \tag{9}$$

The bias and mean square error of $\bar{y}_{BT}^{*'}$, are given by

$$B(\bar{y}_{BT}^{*'}) \cong \frac{1}{\bar{X}} \left( \frac{3R'B^{*'}}{8} - \frac{C^{*'}}{2} \right) \tag{10}$$

and

$$MSE(\bar{y}_{BT}^{*'}) \cong \left[ A^{*'} + \frac{R'^2 B^{*'}}{4} - R' C^{*'} \right]. \tag{11}$$

## 2.5 Singh and Kumar (2010) estimator

Singh and Kumar (2010) estimator under simple random sampling, is given by

$$\bar{y}_{SK}^* = \bar{z}^* \left( \frac{\bar{X}}{\bar{x}^*} \right)^2. \tag{12}$$

The bias and mean square error of $\bar{y}_{SK}^{*'}$, are given by

$$B(\bar{y}_{SK}^{*'}) \cong \frac{1}{\bar{X}} \left( 3R'B^{*'} - 2C^{*'} \right) \tag{13}$$

and

$$MSE(\bar{y}_{SK}^{*'}) \cong [A^{*'} + 4R'^2 B^{*'} - 4R' C^{*'}]. \tag{14}$$

## 2.6 Difference estimator

The difference estimator under simple random sampling, is given by

$$\bar{y}_D^{*'} = [\bar{z}^* + d^{*'}(\bar{X} - \bar{x}^{*'})], \tag{15}$$

where $\bar{x}'^* = \frac{N\bar{X} - n\bar{x}^*}{N-n}$ and $d^{*\prime}$ is a constant.

The minimum variance of $\bar{y}_D^{*'}$, is given by

$$Var(\bar{y}_D^{*'})_{min} = \left[ A^{*'} - \frac{C^{*'2}}{B^{*'}} \right]. \tag{16}$$

The optimum value of $d$ is $d_{(opt)} = -\frac{C^{*'}}{tB^{*'}}$, where $t = \frac{n}{N-n}$.

## 2.7 Azeem and Hanif (2017) estimator

Azeem and Hanif (2017) estimator under simple random sampling, is given by

$$\bar{y}_{AH}^{*'} = \bar{z}^* \left( \frac{\bar{x}^{*'}}{\bar{X}} \right) \left( \frac{\bar{x}^{*'}}{\bar{x}^*} \right). \tag{17}$$

The bias and MSE of $\bar{y}_{AH}^{*'}$, are given by

$$B(\bar{y}_{AH}^{*'}) \cong \frac{1}{\bar{X}} [t^2 R' B^{*'} - qC^{*'}] \tag{18}$$

and

$$MSE(\bar{y}_{AH}^{*'})) \cong [A^{*'} + q^2 R^{'2} B^{*'} - 2q R' C^{*'}], \qquad (19)$$

where $q = \frac{N+n}{N-n}$.

## 3 Proposed generalized class of estimators

We propose a generalized class of estimators for the population mean for a sensitive variable considering the problem of measurement error and non-response simultaneously. Measurement error and non-response are present on both the study variable and the auxiliary variable. The proposed estimator is given by

$$\bar{y}_{GP}^{*'} = \left\{ m_1 \bar{z}^* \left\{ \frac{\bar{X}}{\bar{x}'^*} \right\}^{\alpha_1} + m_2 (\bar{X} - \bar{x}'^*) \left\{ \frac{\bar{X}}{\bar{x}'^*} \right\}^{\alpha_2} + m_3 (\bar{R}_x - \bar{r}'^*_x) \left\{ \frac{\bar{X}}{\bar{x}'^*} \right\}^{\alpha_3} \right\}$$
$$exp(1 - \alpha_0) \left( \frac{\bar{X} - \bar{x}'^*}{\bar{X} + \bar{x}'^*} \right), \qquad (20)$$

where, $m_1$, $m_2$ and $m_3$ are constants whose values are to be determined, and $\alpha_r (r = 0, 1, 2, 3)$ are the scalars, chosen arbitrarily. For obtaining the bias and mean square error, we assume that

$\delta_Z^* = \sum_{i=1}^{n} (Z_i^* - \bar{Z}), \delta_Q^* = \sum_{i=1}^{n} Q_i^*,$
$\delta_X^* = \sum_{i=1}^{n} (X_i^* - \bar{X}), \delta_V^* = \sum_{i=1}^{n} V_i^*,$
$\delta_{R_x}^* = \sum_{i=1}^{n} (R_{x_i}^* - \bar{R}_x), \delta_T^* = \sum_{i=1}^{n} T_i^*.$
Adding $\delta_Y^*$ and $\delta_U^*$, we get $\delta_Z^* + \delta_U^* = \sum_{i=1}^{n} (Z_i^* - \bar{Z}) + \sum_{i=1}^{n} U_i^*.$
Dividing both sides by $n$, and then simplifying, we get

$$\bar{z}^* = \bar{Y} + \frac{1}{n} (\delta_Z^* + \delta_Q^*), \qquad (21)$$

$$\bar{x}^* = \bar{X} + \frac{1}{n} (\delta_X^* + \delta_V^*) \qquad (22)$$

and

$$\bar{r}_x^* = \bar{R}_x + \frac{1}{n} (\delta_{R_x}^* + \delta_T^*). \qquad (23)$$

Further

$E\left( \frac{\delta_Z^* + \delta_Q^*}{n} \right)^2 = \lambda_2 (S_Z^2 + S_Q^2) + \theta(S_{Z(2)}^2 + S_{Q(2)}^2) = A^{*'},$

$E\left( \frac{\delta_X^* + \delta_V^*}{n} \right)^2 = \lambda_2 (S_X^2 + S_V^2) + \theta(S_{X(2)}^2 + S_{V(2)}^2) = B^{*'},$

$E\left( \frac{\delta_{R_x}^* + \delta_T^*}{n} \right)^2 = \lambda_2 (S_{R_x}^2 + S_T^2) + \theta(S_{R_x(2)}^2 + S_{T(2)}^2) = D^{*'},$

$E\left( \frac{\delta_Z^* + \delta_Q^*}{n} \right)\left( \frac{\delta_X^* + \delta_V^*}{n} \right) = \lambda_2 \rho_{ZX} S_Z S_X + \theta\rho_{ZX(2)} S_{Z(2)} S_{X(2)} = C^{*'},$

$E\left( \frac{\delta_Z^* + \delta_U^*}{n} \right)\left( \frac{\delta_{R_x}^* + \delta_T^*}{n} \right) = \lambda_2 \rho_{ZR_x} S_Z S_{R_x} + \theta\rho_{ZR_x(2)} S_{Z(2)} S_{R_x(2)} = E^{*'},$

$E\left( \frac{\delta_X^* + \delta_V^*}{n} \right)\left( \frac{\delta_{R_x}^* + \delta_T^*}{n} \right) = \lambda_2 \rho_{XR_x} S_X S_{R_x} + \theta\rho_{XR_x(2)} S_{X(2)} S_{R_x(2)} = F^{*'}.$

On simplifying, we get

$$\bar{y}_{GP}^{*'} = m_1\left(\bar{Z} + W_Z + e^*R'tW_X + \frac{f^*t^2R'W_X^2}{\bar{X}} + e^*t\frac{W_XW_Z}{\bar{X}}\right) + m_2\left(tW_X + d^*t^2\frac{W_X^2}{\bar{X}}\right)$$
$$+ m_3\left(tW_{R_x} + c^*t\frac{W_XW_{R_x}}{\bar{X}} + b^*t^2\frac{W_{R_x}^2}{\bar{R}_x}\right), \tag{24}$$

where

$b^* = \alpha_3,$

$c^* = \frac{1-\alpha_0}{2},$

$d^* = \alpha_2 + \frac{1-\alpha_0}{2},$

$e^* = \alpha_1 + \frac{1-\alpha_0}{2},$ and

$f^* = \frac{\alpha_0^2 - 4\alpha_0 + 3}{8} + \frac{\alpha_1(2-\alpha_0+\alpha_1)}{2}.$

$W_Z = \frac{\delta_Z^* + \delta_Q^*}{n}, W_X = \frac{\delta_X^* + \delta_V^*}{n}$ and $W_{R_x} = \frac{\delta_{R_x}^* + \delta_T^*}{n}.$

Simplifying further, and ignoring error terms of power greater than two, we have

$$\bar{y}_{GP}^{*'} - \bar{Z} = (m_1 - 1)\bar{Z} + m_1\left(W_Z + e^*R'tW_X + f^*t^2R'W_X^2 + e^*t\frac{W_XW_Z}{\bar{X}}\right)$$
$$+ m_2\left(tW_X + d^*t^2\frac{W_X^2}{\bar{X}}\right) + m_3\left(tW_{R_x} + c^*t\frac{W_XW_{R_x}}{\bar{X}} + b^*t^2\frac{W_{R_x}^2}{\bar{R}_x}\right). \tag{25}$$

Using Eq (25), the bias of $\bar{y}_{GP}^{*'}$, to first order of approximation, is given by

$$B(\bar{y}_{GP}^{*'}) \cong \left[(m_1-1)\bar{Z} + m_1\left(\frac{f^*t^2R'B^{*'}}{\bar{X}} + \frac{e^*tC^{*'}}{\bar{X}}\right)\right.$$
$$\left. + m_2\left(\frac{d^*t^2B^{*'}}{\bar{X}}\right) + m_3\left(\frac{c^*tF^{*'}}{\bar{X}} + \frac{b^*t^2D^{*'}}{\bar{R}_x}\right)\right]. \tag{26}$$

Squaring both sides of Eq (25), and keeping the terms up to power two in errors, and then taking expectations, the mean square error of $\bar{y}_{GP}^{*'}$ is given by

$$MSE(\bar{y}_{GP}^{*'}) \cong [\bar{Z}^2 + m_1^2(\bar{Z}^2 + A^{*'} + e^{*2}t^2R'^2B^{*'} + 4e^*tR'C^{*'} + 2f^*t^2R'^2B^{*'}) + m_2^2t^2B^{*'}$$
$$+ 2m_1m_2(tC^{*'} + t^2R'B^{*'}(e^* + d^*)) - 2m_1(\bar{Z}^2 + e^*tR'C^{*'} + f^*t^2R'^2B^{*'})$$
$$- 2m_2d^*t^2R'B^{*'} + m_3^2t^2D^{*'} + 2m_1m_3(c^*tR'F^{*'} + e^*t^2R'F^{*'} + tE^{*'} + b^*t^2R_1'D^{*'})$$
$$+ 2m_2m_3t^2F^{*'} - 2m_3(c^*tR'F^{*'} + b^*t^2R_1'D^{*'})],$$

where $R_1' = \frac{\bar{Z}}{\bar{R}_x}.$

The above equation can be written as

$$MSE(\bar{y}_{GP}^{*'}) \cong [\bar{Z}^2 + m_1^2A_1^{*'} + m_2^2B_1^{*'} + 2m_1m_2C_1 - 2m_1D_1^{*'}$$
$$- 2m_2E_1 + m_3^2F_1^{*'} + 2m_1m_3G_1^{*'} + 2m_2m_3H_1^{*'} - 2m_3I_1^{*'}], \tag{27}$$

where,

$A_1^{*'} = \bar{Z}^2 + A^{*'} + e^{*2}t^2R'^2B^{*'} + 4e^*tR'C^{*'} + 2f^*t^2R'^2B^{*'},$

$B_1^{*'} = t^2B^{*'},$

$C_1^{*'} = tC^{*'} + t^2R'B^{*'}(e^* + d^*),$

$$D_1^{*'} = \bar{Z}^2 + e^* t R' C^{*'} + f^* t^2 R'^2 B^{*'},$$
$$E_1^{*'} = d^* t^2 R' B^{*'},$$
$$F_1^{*'} = t^2 D^{*'},$$
$$G_1^{*'} = c^* t R' F^{*'} + e^* t^2 R' F^{*'} + tE + b^* t^2 R_1' D^{*'},$$
$$H_1^{*'} = t^2 F^{*'},$$
$$I_1^{*'} = c^* t R' F^{*'} + b^* t^2 R_1' D^{*'}.$$

For finding the optimal values of $m_1$, $m_2$ and $m_3$, we differentiate Eq (27) with respect to $m_1$, $m_2$ and $m_3$ respectively. The optimal values are given by

$$m_{1(opt)} = \frac{B_1^{*'} D_1^{*'} F_1^{*'} - C_1^{*'} E_1^{*'} F_1^{*'} + E_1^{*'} G_1^{*'} H_1^{*'} - D_1^{*'} H_1^{*'2} - B_1^{*'} G_1^{*'} I_1^{*'} + C_1^{*'} H_1^{*'} I_1^{*'}}{A_1^{*'} B_1^{*'} F_1^{*'} - C_1^{*'2} F_1^{*'} + 2 C_1^{*'} G_1^{*'} H_1^{*'} - A_1^{*'} H_1^{*'2},}$$

$$m_{2(opt)} = \frac{A_1^{*'} E_1^{*'} F_1^{*'} - C_1^{*'} D_1^{*'} F_1^{*'} - E_1^{*'} G_1^{*'2} + D_1^{*'} G_1^{*'} H_1^{*'} + C_1^{*'} G_1^{*'} I_1^{*'} - A_1^{*'} H_1^{*'} I_1^{*'}}{A_1^{*'} B_1^{*'} F_1^{*'} - C_1^{*'2} F_1^{*'} + 2 C_1^{*'} G_1^{*'} H_1^{*'} - A_1^{*'} H_1^{*'2}},$$

and

$$m_{3(opt)} = \frac{C_1^{*'} E_1^{*'} G_1^{*'} - B_1^{*'} D_1^{*'} G_1^{*'} + C_1^{*'} D_1^{*'} H_1^{*'} - A_1^{*'} E_1^{*'} H_1^{*'} + A_1^{*'} B_1^{*'} I_1^{*'} - C_1^{*'2} I_1^{*'}}{A_1^{*'} B_1^{*'} F_1^{*'} - C_1^{*'2} F_1^{*'} + 2 C_1^{*'} G_1^{*'} H_1^{*'} - A_1^{*'} H_1^{*'2}}$$

Substituting these optimum values in Eq (27), we get the minimum mean square error of $\bar{y}_{GP}^{*'}$, as

$$MSE(\bar{y}_{GP}^{*'})_{min} \cong \left[ \bar{Z}^2 - \frac{L_1^{*'}}{L_2^{*'}} \right], \tag{28}$$

where

$$L_1^{*'} = A_1^{*'} E_1^{*'2} F_1^{*'} - 2 C_1^{*'} D_1^{*'} E_1^{*'} F_1^{*'} - E_1^{*'2} G_1^{*'2} + 2 D_1^{*'} E_1^{*'} G_1^{*'} H_1^{*'} - D_1^{*'2} H_1^{*'2} + 2 C_1^{*'} E_1^{*'} G_1^{*'} I_1^{*'} + 2 C_1^{*'} D_1^{*'} H_1^{*'} I_1^{*'} - 2 A_1^{*'} E_1^{*'} H_1^{*'} I_1^{*'} - C_1^{*'2} I_1^{*'2} + B_1^{*'} D_1^{*'2} F_1^{*'} - 2 B_1^{*'} D_1^{*'} G_1^{*'} I_1^{*'} + B_1^{*'} A_1^{*'} I_1^{*'2}$$

and

$$L_2^{*'} = A_1^{*'} B_1^{*'} F_1^{*'} - C_1^{*'2} F_1^{*'} + 2 C_1^{*'} G_1^{*'} H_1^{*'} - A_1^{*'} H_1^{*'2} - B_1^{*'} G_1^{*'2}.$$

## 3.1 Specific members of generalized proposed class of estimators $\bar{y}_{GP}^{*'}$ for different choices of $(\alpha_0, \alpha_1, \alpha_2, \alpha_3, m_1, m_2, m_3)$

We consider the following members of the class of estimators $\bar{y}_{GP}^{*'}$ by choosing different values of $\alpha_0$, $\alpha_1$, $\alpha_2$, $\alpha_3$, $m_1$, $m_2$ and $m_3$.

1. For $\alpha_1 = m_2 = m_3 = 0$ and $\alpha_0 = m_1 = 1$ in Eq 20, the generalized proposed class of estimators $\bar{y}_{GP}^{*'}$ reduces to usual mean estimator:

$$\bar{y}_0^{*'} = \bar{z}^*.$$

2. For $\alpha_0 = \alpha_1 = m_1 = 1$ and $m_2 = m_3 = 0$ in Eq 20, the generalized proposed class of estimators $\bar{y}_{GP}^{*'}$ reduces to usual ratio estimator:

$$\bar{y}_R^{*'} = \frac{\bar{z}^*}{\bar{x}^*} \bar{X}.$$

3. For $\alpha_0 = m_1 = 1$, $\alpha_1 = -1$ and $m_2 = m_3 = 0$ in Eq 20, the generalized proposed class of estimators $\bar{y}_{GP}^{*'}$ reduces to usual product estimator:

$$\bar{y}_{Pr}^{*'} = \bar{z}^* \frac{\bar{x}^*}{\bar{X}}.$$

4. For $\alpha_0 = \alpha_1 = m_2 = m_3 = 0$ and $m_1 = 1$ in Eq 20, the generalized proposed class of estimators $\bar{y}_{GP}^{*'}$ reduces to the estimator in Eq 9:

$$\bar{y}_{BT}^{*'} = \bar{z}^* \exp\left(\frac{\bar{X} - \bar{x}^*}{\bar{X} + \bar{x}^*}\right).$$

5. For $\alpha_0 = \alpha_1 = \alpha_2 = 1$, $m_3 = 0$, $m_1 = m_4$ and $m_2 = m_5$ in Eq 20, the generalized proposed class of estimators $\bar{y}_{GP}^{*'}$ reduces to the following estimator.

$$\bar{y}_{GS}^{*'} = m_4 \bar{z}^* \left(\frac{\bar{X}}{\bar{x}^*}\right) + m_5(\bar{X} - \bar{x}^*)\left(\frac{\bar{X}}{\bar{x}^*}\right).$$

6. For $\alpha_0 = m_2 = m_3 = 0$, $\alpha_1 = 2$ and $m_1 = 1$ in Eq 20, the generalized proposed class of estimators $\bar{y}_{GP}^{*'}$ reduces to the estimator in Eq 12:

$$\bar{y}_{SK}^{*'} = \bar{z}^* \left(\frac{\bar{X}}{\bar{x}^*}\right)^2.$$

7. For $\alpha_0 = \alpha_1 = \alpha_2 = 0$, $m_3 = 0$, $m_1 = m_6$ and $m_2 = m_7$ in Eq 20, the generalized proposed class of estimators $\bar{y}_{GP}^{*'}$ reduces to the following estimator:

$$\bar{y}_{GK}^{*'} = m_6 \bar{z}^* + m_7(\bar{X} - \bar{x}^{*'})\exp\left(\frac{\bar{X} - \bar{x}^{*'}}{\bar{X} + \bar{x}^{*'}}\right).$$

8. For $\alpha_1 = \alpha_2 = m_3 = 0$, $\alpha_0 = m_1 = 1$ and $m_2 = d$ in Eq 20, the generalized proposed class of estimators $\bar{y}_{GP}^{*'}$ reduces to difference estimator:

$$\bar{y}_D^{*'} = [\bar{z}^* + d(\bar{X} - \bar{x}^{*'})],$$

9. For $\alpha_0 = \alpha_1 = \alpha_2 = \alpha$, $m_1 = m_8$, $m_2 = m_9$ and $m_3 = 0$ in Eq 20, the generalized proposed estimator $\bar{y}_{GP}^{*'}$ reduces to another form of proposed estimator.

$$\bar{y}_{P1}^{*'} = \left[\left\{m_8\bar{z}^* + m_9(\bar{X} - \bar{x}^{*'})\right\}\left\{\frac{\bar{X}}{\bar{x}^*}\right\}^\alpha exp(1 - \alpha)\left(\frac{\bar{X} - \bar{x}^{*'}}{\bar{X} + \bar{x}^{*'}}\right)\right], \tag{29}$$

## 3.2 Efficiency comparison

The efficiency comparison of $\bar{y}_{HH}^{*'}, \bar{y}_R^{*'}, \bar{y}_{Pr}^{*'}, \bar{y}_{BT}^{*'}, \bar{y}_{SK}^{*'}, \bar{y}_D^{*'}$ and $\bar{y}_{AH}^{*'}$ with respect to $\bar{y}_{GP}^{*'}$ are given by:

**Condition (1)**

From Eqs (2) and (28),

$MSE(\bar{y}_{GP}^{*'})_{min} < Var(\bar{y}_{HH}^{*'})$ if

$\bar{Z}^2 - \left[\frac{L_1^{*'}}{L_2^{*'}}\right] - A^{*'} < 0$

**Condition (2)**

From Eqs (5) and (28),

$MSE(\bar{y}_{GP}^{*'})_{min} < MSE(\bar{y}_{R}^{*'})$ if

$$\bar{Z}^2 - \left[\frac{L_1^{*'}}{L_2^{*'}}\right] - [A^{*'} + R'^2 B^{*'} - 2R'C^{*'}] < 0$$

**Condition (3)**

From Eqs (8) and (28),

$MSE(\bar{y}_{GP}^{*'})_{min} < MSE(\bar{y}_{Pr}^{*'})$ if

$$\bar{Z}^2 - \left[\frac{L_1^{*'}}{L_2^{*'}}\right] - [A^{*'} + R'^2 B^{*'} + 2R'C^{*'}] < 0$$

**Condition (4)**

From Eqs (11) and (28),

$MSE(\bar{y}_{GP}^{*'})_{min} < MSE(\bar{y}_{BT}^{*'})$ if

$$\bar{Z}^2 - \left[\frac{L_1^{*'}}{L_2^{*'}}\right] - \left[A^{*'} + \frac{R'^2 B^{*'}}{4} - R'C^{*'}\right] < 0$$

**Condition (5)**

From Eqs (14) and (28),

$MSE(\bar{y}_{GP}^{*'})_{min} < MSE(\bar{y}_{SK}^{*'})$ if

$$\bar{Z}^2 - \left[\frac{L_1^{*'}}{L_2^{*'}}\right] - [A^{*'} + 4R'^2 B^{*'} - 4R'C^{*'}] < 0$$

**Condition (6)**

From Eqs (16) and (28),

$MSE(\bar{y}_{GP}^{*'})_{min} < Var(\bar{y}_{D}^{*'})_{min}$ if

$$\bar{Z}^2 - \left[\frac{L_1^{*'}}{L_2^{*'}}\right] - \left[A^{*'} - \frac{C^2}{B^{*'}}\right] < 0$$

**Condition (7)**

From Eqs (19) and (28),

$MSE(\bar{y}_{GP}^{*'})_{min} < MSE(\bar{y}_{AH}^{*'})$ if

$$\bar{Z}^2 - \left[\frac{L_1^{*'}}{L_2^{*'}}\right] - [A^{*'} + q^2 R'^2 B^{*'} - 2qR'C^{*'}] < 0$$

The proposed class of estimators $\bar{y}_{GP}^{*'}$ is more efficient than the competing estimators when Conditions (1) to (7) are satisfied. Table 1 shows that all the conditions are satisfied.

## 4 Numerical results

In this section three populations are generated for simulation study and three real data sets are used. The results are given in Tables 2–7 (simulated data) and Tables 9–14 (real data).

### 4.1 Simulation study

We have generated three populations from a normal distribution by using R language program. In Tables 2–7, we can see that the MSE for the generalized proposed estimator is minimum, both with and without measurement error. The value for the bias (in brackets) of the estimators are also given in Tables 2–7.

**Population I**.

$X = N(5, 10)$, $Y = X + N(0, 1)$, $y = Y + N(1, 3)$, $x = X + N(1, 3)$, $N = 5000$, $\bar{Y} = 4.927167$, $\bar{X} = 4.924306$, $\bar{R}_x = 2500.5$, $S_Y^2 = 102.0075$, $S_X^2 = 101.4117$, $S_{R_x}^2 = 2083750$, $S_U^2 = 8.862114$, $S_V^2 = 9.001304$, $S_T^2 = 0.250076$, $\rho_{YX} = 0.995059$, $\rho_{YR_x} = 0.008771$, $\rho_{XR_x} = 0.005563$.

**Table 1. Conditional values for the efficiency comparison for population(I-VI).**

| Conditions | | Populations | | | | | |
|---|---|---|---|---|---|---|---|
| | | I | II | III | IV | V | VI |
| With ME | 1 | −0.107129 | −0.107769 | −0.023380 | −0.005017 | −0.425301 | −0.418561 |
| | 2 | −0.002396 | −0.001124 | −0.003177 | −0.000151 | −2.686431 | −4.785247 |
| | 3 | −0.460724 | −0.475521 | −0.130722 | −0.022673 | −3.212175 | −4.832798 |
| | 4 | −0.023655 | −0.021808 | −0.002386 | −0.000986 | −0.924866 | −1.504289 |
| | 5 | −0.146525 | −0.155586 | −0.070112 | −0.008074 | −9.995565 | −17.93285 |
| | 6 | −0.001617 | −0.000029 | −0.000044 | −0.000060 | −0.418457 | −0.418529 |
| | 7 | −0.027958 | −0.029178 | −0.018401 | −0.001537 | −6.123849 | −11.00540 |
| Without ME | 1 | −0.115168 | −0.115461 | −0.028897 | −0.005075 | −0.417171 | −0.368190 |
| | 2 | −0.000010 | −0.000022 | −0.000310 | −0.000089 | −1.770229 | −2.652350 |
| | 3 | −0.458333 | −0.462384 | −0.127856 | −0.022610 | −2.295974 | −3.358644 |
| | 4 | −0.029086 | −0.028805 | −0.005807 | −0.001014 | −0.689718 | −0.850944 |
| | 5 | −0.112845 | −0.116059 | −0.042094 | −0.007650 | −6.355147 | −10.211124 |
| | 6 | −0.000006 | −0.000021 | 0.000000 | −0.000023 | −0.406481 | −0.356371 |
| | 7 | −0.016107 | −0.016930 | −0.007930 | −0.001371 | −3.918410 | −6.200481 |

**Population II.**

$X = N(5, 10)$, $Y = X + N(0, 1)$, $y = Y + N(2, 3)$, $x = X + N(2, 3)$, $N = 5000$, $\bar{Y} = 4.730993$, $\bar{X} = 4.741928$, $\bar{R}_x = 2500.5$, $S_Y^2 = 101.2633$, $S_X^2 = 100.2288$, $S_{R_x}^2 = 2083750$, $S_U^2 = 9.1025$, $S_V^2 = 9.052019$, $S_T^2 = 0.25487$, $\rho_{YX} = 0.995187$, $\rho_{YR_x} = -0.018591$, $\rho_{XR_x} = -0.019483$.

**Population III.**

$X = N(5, 10)$, $Y = X + N(0, 1)$, $y = Y + N(1, 2.5)$, $x = X + N(1, 2.5)$, $N = 5000$, $\bar{Y} = 2.14160$, $\bar{X} = 1.943369$, $\bar{R}_x = 2500.5$, $S_Z^2 = 25.38513$, $S_X^2 = 24.50418$, $S_{R_x}^2 = 2083750$, $S_U^2 = 6.040431$, $S_V^2 = 6.224441$, $S_T^2 = 0.25487$, $\rho_{ZX} = 0.9749021$, $\rho_{ZR_x} = 0.003505$, $\rho_{XR_x} = 0.029275$.

Tables 2–7 show that the generalized class of proposed estimators $\bar{y}_{GP}^{*'}$ performs better than all other estimators for both with and without measurement errors. The values of the absolute

**Table 2. Mean squared error and |Bias| (in brackets) values for different estimators for Population I with measurement error.**

| Estimators | 10% non-response | | | 20% non-response | | |
|---|---|---|---|---|---|---|
| | g | | | g | | |
| | 2 | 4 | 8 | 2 | 4 | 8 |
| $\bar{y}_{HH}^{*'}$ | 0.128362 | 0.157619 | 0.216131 | 0.145653 | 0.202286 | 0.315553 |
| $\bar{y}_{R}^{*'}$ | 0.023629 (0.001954) | 0.028780 (0.002331) | 0.039080 (0.003085) | 0.026448 (0.002423) | 0.036952 (0.003412) | 0.057959 (0.005389) |
| $\bar{y}_{Pr}^{*'}$ | 0.481957 (0.022740) | 0.591127 (0.027902) | 0.809466 (0.038224) | 0.552584 (0.025858) | 0.767722 (0.035916) | 1.197998 (0.056031) |
| $\bar{y}_{BT}^{*'}$ | 0.044888 (0.054066) | 0.055115 (0.066978) | 0.075570 (0.092803) | 0.050085 (0.059786) | 0.069606 (0.082597) | 0.108649 (0.128217) |
| $\bar{y}_{SK}^{*'}$ | 0.167758 (0.028605) | 0.204610 (0.034896) | 0.278313 (0.047480) | 0.194969 (0.033130) | 0.271718 (0.046153) | 0.425215 (0.072200) |
| $\bar{y}_{D}^{*'}$ | 0.022850 | 0.027874 | (0.037919 | 0.025392 | 0.035446 | 0.055553 |
| $\bar{y}_{AH}^{*'}$ | 0.049191 (0.030508) | 0.059838 (0.037434) | 0.081131 (0.051287) | 0.056720 (0.034683) | 0.079209 (0.048171) | 0.124187 (0.075148) |
| $\alpha = 0, \bar{y}_{P}^{*'}$ | 0.022828 (0.004530) | 0.027841 (0.005525) | 0.037860 (0.007514) | 0.025365 (0.004986) | 0.035395 (0.006958) | 0.055428 (0.010896) |
| $\alpha = 1, \bar{y}_{P1}^{*'}$ | 0.022829 (0.004530) | 0.027843 (0.005526) | 0.037863 (0.007514) | 0.025367 (0.004987) | 0.035397 (0.006958) | 0.055434 (0.010898) |
| $\alpha = -1, \bar{y}_{P1}^{*'}$ | 0.022830 (0.004531) | 0.027844 (0.005527) | 0.037864 (0.007515) | 0.025368 (0.004988) | 0.035398 (0.006959) | 0.055435 (0.010899) |
| $\alpha_r = 1, r = 0, 1, 2, 3\ \bar{y}_{GP}^{*'}$ | **0.021233** (0.008037) | 0.027317 (0.011036) | 0.037453 (0.015455) | 0.023755 (0.009102) | 0.034801 (0.0141924) | 0.054783 (0.022759) |
| $\alpha_0 = 0, \alpha_{1,2,3} = 1\ \bar{y}_{GP}^{*'}$ | 0.021305 (0.004318) | 0.027434 (0.006204) | 0.037673 (0.008838) | 0.023845 (0.004930) | 0.034992 (0.008020) | 0.055255 (0.013061) |

**Table 3. Mean squared error and |*Bias*| (in brackets) values for different estimators for Population I without measurement error.**

| Estimators | 10% non-response | | | 20% non-response | | |
|---|---|---|---|---|---|---|
| | g | | | g | | |
| | 2 | 4 | 8 | 2 | 4 | 8 |
| $\bar{y}_{HH}^{*'}$ | 0.117513 | 0.144220 | 0.197633 | 0.133610 | 0.185307 | 0.288702 |
| $\bar{y}_{R}^{*'}$ | 0.002355 (0.000115) | 0.002859 (0.000154) | 0.003862 (0.000233) | 0.002612 (0.000103) | 0.003609 (0.000194) | 0.005618 (0.000377) |
| $\bar{y}_{Pr}^{*'}$ | 0.460678 (0.022740) | 0.565200 (0.027902) | 0.774244 (0.038224) | 0.528737 (0.025858) | 0.734374 (0.035916) | 1.145649 (0.056031) |
| $\bar{y}_{BT}^{*'}$ | 0.031431 (0.073960) | 0.038584 (0.090874) | 0.052891 (0.124703) | 0.035091 (0.082179) | 0.048535 (0.113649) | 0.075424 (0.176587) |
| $\bar{y}_{SK}^{*'}$ | 0.115190 (0.022395) | 0.141100 (0.027437) | 0.192919 (0.037522) | 0.135710 (0.026168) | 0.189265 (0.036499) | 0.296374 (0.057162) |
| $\bar{y}_{D}^{*'}$ | 0.002351 | 0.002855 | 0.003856 | 0.002606 | 0.003605 | 0.005599 |
| $\bar{y}_{AH}^{*'}$ | 0.018452 (0.030583) | 0.022548 (0.037524) | 0.030740 (0.051407) | 0.022166 (0.034767) | 0.031015 (0.048288) | 0.048711 (0.075329) |
| $\alpha=0,\bar{y}_{P1}^{*'}$ | 0.002348 (0.000465) | 0.002849 (0.000565) | 0.003851 (0.000764) | 0.002601 (0.000511) | 0.003600 (0.000707) | 0.005595 (0.001100) |
| $\alpha=1,\bar{y}_{P1}^{*'}$ | 0.002349 (0.000465) | 0.002850 (0.000565) | 0.003852 (0.000764) | 0.002602 (0.000512) | 0.003601 (0.000707) | 0.005596 (0.001100) |
| $\alpha=-1,\bar{y}_{P1}^{*'}$ | 0.002350 (0.000465) | 0.002851 (0.000565) | 0.003853 (0.000764) | 0.002603 (0.000513) | 0.003602 (0.000707) | 0.005596 (0.001100) |
| $\alpha_r=1, r=0,1,2,3\ \bar{y}_{GP}^{*'}$ | **0.002345** (0.001451) | 0.002846 (0.001771) | 0.003846 (0.002411) | 0.002597 (0.001616) | 0.003595 (0.002244) | 0.005587 (0.003499) |
| $\alpha_0=0, \alpha_{1,2,3}=1\ \bar{y}_{GP}^{*'}$ | 0.002347 (0.001409) | 0.002848 (0.001720) | 0.003850 (0.002341) | 0.002599 (0.001580) | 0.003598 (0.002195) | 0.005591 (0.003423) |

biases are given in brackets. Table 2 shows that generalized proposed estimator performs better than other estimators. The MSE for the generalized proposed estimator, when $\alpha_r = 1$, $r = 0$, 1, 2, 3 is 0.021233 for 10% non-response rate. When the non-response rate increases to 20%, the MSE for generalized proposed estimator increases to 0.023755. It is also observed that $\bar{y}_R^{*'}$ is less biased and $\bar{y}_{BT}^{*'}$ is most biased among all estimators. Table 3 shows the same pattern of results.

Table 4 shows that generalized proposed estimators performs better than other estimators. The MSE for the generalized proposed estimator, when $\alpha_r = 1$, $r = 0$, 1, 2, 3 is 0.021796 for 10% non-response rate. When the non-response rate becomes 20%, the MSE for generalized proposed estimator increases to 0.024110. It is also observed that $\bar{y}_R^{*'}$ is less biased and $\bar{y}_{BT}^{*'}$ is most biased among all estimators. Table 5 shows the same pattern of results.

**Table 4. Mean squared error and |*Bias*| (in brackets) values for different estimators for Population II with measurement error.**

| Estimators | 10% non-response | | | 20% non-response | | |
|---|---|---|---|---|---|---|
| | g | | | g | | |
| | 2 | 4 | 8 | 2 | 4 | 8 |
| $\bar{y}_{HH}^{*'}$ | 0.129565 | 0.157215 | 0.212515 | 0.140944 | 0.196138 | 0.306526 |
| $\bar{y}_{R}^{*'}$ | 0.022920 (0.002412) | 0.027877 (0.003071) | 0.037791 (0.004391) | 0.024956 (0.002102) | 0.033062 (0.002779) | 0.049274 (0.004131) |
| $\bar{y}_{Pr}^{*'}$ | 0.497317 (0.023931) | 0.606125 (0.029170) | 0.823741 (0.039647) | 0.531056 (0.025251) | 0.741067 (0.035325) | 1.161088 (0.055473) |
| $\bar{y}_{BT}^{*'}$ | 0.043604 (0.049617) | 0.052599 (0.059304) | 0.070590 (0.07867) | 0.048685 (0.060175) | 0.066868 (0.085733) | 0.103236 (0.136849) |
| $\bar{y}_{SK}^{*'}$ | 0.177382 (0.0311678) | 0.218111 (0.038386) | 0.299570 (0.052822) | 0.183091 (0.031560) | 0.251838 (0.043662) | 0.389331 (0.067867) |
| $\bar{y}_{D}^{*'}$ | 0.021825 | 0.026426 | 0.035621 | 0.024146 | 0.032046 | 0.047839 |
| $\bar{y}_{AH}^{*'}$ | 0.050974 (0.032092) | 0.062665 (0.039112) | 0.086047 (0.053154) | 0.052875 (0.033878) | 0.071379 (0.047400) | 0.108388 (0.074443) |
| $\alpha=0,\bar{y}_{P1}^{*'}$ | 0.021805 (0.004399) | 0.026396 (0.005322) | 0.035566 (0.007176) | 0.024121 (0.004813) | 0.032003 (0.006381) | 0.047742 (0.009528) |
| $\alpha=1,\bar{y}_{P1}^{*'}$ | 0.021806 (0.004400) | 0.026397 (0.005326) | 0.035568 (0.007177) | 0.024122 (0.004814) | 0.032005 (0.006385) | 0.047747 (0.009529) |
| $\alpha=-1,\bar{y}_{P1}^{*'}$ | 0.021807 (0.004401) | 0.026398 (0.005327) | 0.035569 (0.007178) | 0.024123 (0.004815) | 0.032006 (0.006387) | 0.047748 (0.009529) |
| $\alpha_r=1, r=0,1,2,3\ \bar{y}_{GP}^{*'}$ | **0.021796** (0.005713) | 0.026384 (0.006931) | 0.035544 (0.009363) | 0.024110 (0.006008) | 0.031983 (0.008057) | 0.047698 (0.012147) |
| $\alpha_0=0, \alpha_{1,2,3}=1\ \bar{y}_{GP}^{*'}$ | 0.021801 (0.005609) | 0.026391 (0.006804) | 0.035562 (0.009191) | 0.024119 (0.005904) | 0.032000 (0.007918) | 0.047732 (0.011939) |

**Table 5. Mean squared error and |Bias| (in brackets) values for different estimators for Population II without measurement error.**

| Estimators | 10% non-response | | | 20% non-response | | |
|---|---|---|---|---|---|---|
| | g | | | g | | |
| | 2 | 4 | 8 | 2 | 4 | 8 |
| $\bar{y}_{HH}^{*'}$ | 0.117648 | 0.143868 | 0.196307 | 0.131505 | 0.186699 | 0.297087 |
| $\bar{y}_{R}^{*'}$ | 0.002209 (0.000030) | 0.002484 (0.000036) | 0.003040 (0.000048) | 0.006076 (0.000218) | 0.014182 (0.000895) | 0.030394 (0.002247) |
| $\bar{y}_{Pr}^{*'}$ | 0.464571 (0.023683) | 0.568731 (0.029004) | 0.777050 (0.039646) | 0.512176 (0.025251) | 0.722187 (0.035325) | 1.142209 (0.055473) |
| $\bar{y}_{BT}^{*'}$ | 0.030992 (0.070114) | 0.037741 (0.085876) | 0.051239 (0.117399) | 0.036885 (0.078131) | 0.055069 (0.103689) | 0.091436 (0.154804) |
| $\bar{y}_{SK}^{*'}$ | 0.118246 (0.023775) | 0.144581 (0.029114) | 0.197250 (0.039791) | 0.135890 (0.025908) | 0.204637 (0.038010) | 0.342130 (0.062215) |
| $\bar{y}_{D}^{*'}$ | 0.002208 | 0.002484 | 0.003040 | 0.006067 | 0.014071 | 0.029955 |
| $\bar{y}_{AH}^{*'}$ | 0.019117 (0.031845) | 0.023190 (0.039000) | 0.031336 (0.053310) | 0.025432 (0.033947) | 0.043937 (0.047468) | 0.080946 (0.074512) |
| $\alpha=0, \bar{y}_{P1}^{*'}$ | 0.002205 (0.000451) | 0.002482 (0.000508) | 0.003038 (0.000620) | 0.006065 (0.001209) | 0.014062 (0.002806) | 0.029916 (0.005969) |
| $\alpha=1, \bar{y}_{P1}^{*'}$ | 0.002206 (0.000452) | 0.002483 (0.000509) | 0.003039 (0.000621) | 0.006066 (0.001210) | 0.014063 (0.002807) | 0.029917 (0.005971) |
| $\alpha=-1, \bar{y}_{P1}^{*'}$ | 0.002207 (0.000453) | 0.002484 (0.000509) | 0.003040 (0.000622) | 0.006067 (0.001211) | 0.014064 (0.002808) | 0.029920 (0.005972) |
| $\alpha_r=1, r=0,1,2,3\ \bar{y}_{GP}^{*'}$ | **0.002187** (0.001783) | 0.002465 (0.001499) | 0.003021 (0.000932) | 0.006039 (0.002194) | 0.014029 (0.005955) | 0.029775 (0.013420) |
| $\alpha_0=0, \alpha_{1,2,3}=1\ \bar{y}_{GP}^{*'}$ | 0.002205 (0.001050) | 0.002482 (0.001210) | 0.003036 (0.001530) | 0.006057 (0.004432) | 0.014054 (0.006861) | 0.029908 (0.011721) |

Table 6 also shows that the generalized proposed estimator performs better than other estimators. The MSE for the generalized proposed estimator, when $\alpha_r = 1$, $r = 0, 1, 2, 3$ is 0.014048 for 10% non-response rate. When the non-response rate becomes 20%, the MSE for generalized proposed estimator increases to 0.015831. It is also observed that $\bar{y}_{R}^{*'}$ is less biased and $\bar{y}_{SK}^{*'}$ is most biased among all estimators. Table 7 shows the same pattern of results.

Through the simulation study, it is concluded that the generalized proposed class of estimators perform better as compared to the other existing estimators. For 10% non-response rate the MSE is minimum.

**Table 6. Mean squared error and |Bias| (in brackets) values for different estimators for Population III with measurement error.**

| Estimators | 10% non-response | | | 20% non-response | | |
|---|---|---|---|---|---|---|
| | g | | | g | | |
| | 2 | 4 | 8 | 2 | 4 | 8 |
| $\bar{y}_{HH}^{*'}$ | 0.037428 | 0.045788 | 0.062507 | 0.042313 | 0.059461 | 0.093758 |
| $\bar{y}_{R}^{*'}$ | 0.017225 (0.005455) | 0.021164 (0.006618) | 0.029043 (0.008944) | 0.018321 (0.005055) | 0.025872 (0.006964) | 0.040975 (0.010781) |
| $\bar{y}_{Pr}^{*'}$ | 0.144770 (0.014889) | 0.176353 (0.018115) | 0.239517 (0.024569) | 0.156049 (0.016675) | 0.217752 (0.023231) | 0.341157 (0.036344) |
| $\bar{y}_{BT}^{*'}$ | 0.016434 (0.000758) | 0.020233 (0.000893) | 0.027832 (0.001162) | 0.019099 (0.000774) | 0.0270795 (0.001201) | 0.043039 (0.002054) |
| $\bar{y}_{SK}^{*'}$ | 0.084160 (0.031254) | 0.102482 (0.037970) | 0.139126 (0.051401) | 0.084072 (0.031843) | 0.116984 (0.044125) | 0.182809 (0.068689) |
| $\bar{y}_{D}^{*'}$ | 0.014092 | 0.017372 | 0.023931 | 0.015892 | 0.022555 | 0.035882 |
| $\bar{y}_{AH}^{*'}$ | 0.032449 (0.019822) | 0.039650 (0.024119) | 0.054053 (0.032713) | 0.032787 (0.022239) | 0.045877 (0.030986) | 0.072059 (0.048480) |
| $\alpha=0, \bar{y}_{P1}^{*'}$ | 0.014047 (0.006559) | 0.017304 (0.008080) | 0.023804 (0.011115) | 0.015831 (0.007667) | 0.022434 (0.010864) | 0.035575 (0.017229) |
| $\alpha=1, \bar{y}_{P1}^{*'}$ | 0.014048 (0.006560) | 0.017305 (0.008081) | 0.023806 (0.011116) | 0.015832 (0.007668) | 0.022436 (0.010866) | 0.035581 (0.017232) |
| $\alpha=-1, \bar{y}_{P1}^{*'}$ | 0.014049 (0.006560) | 0.017306 (0.008081) | 0.023807 (0.011117) | 0.015833 (0.007669) | 0.022437 (0.010868) | 0.035582 (0.017234) |
| $\alpha_r=1, r=0,1,2,3\ \bar{y}_{GP}^{*'}$ | *0.014048* (0.006550) | 0.017306 (0.008079) | 0.023807 (0.011112) | 0.015831 (0.007667) | 0.022436 (0.010866) | 0.035582 (0.017224) |
| $\alpha_0=0, \alpha_{1,2,3}=1\ \bar{y}_{GP}^{*'}$ | 0.014052 (0.006561) | 0.017311 (0.008083) | 0.023817 (0.011121) | 0.015836 (0.007669) | 0.022445 (0.010870) | 0.035603 (0.017242) |

**Table 7. Mean squared error and |*Bias*| (in brackets) values for different estimators for Population III without measurement error.**

| Estimators | 10% non-response | | | 20% non-response | | |
|---|---|---|---|---|---|---|
| | g | | | g | | |
| | 2 | 4 | 8 | 2 | 4 | 8 |
| $\bar{y}^{*'}_{HH}$ | 0.030001 | 0.036695 | 0.050080 | 0.034053 | 0.047871 | 0.075506 |
| $\bar{y}^{*'}_{R}$ | 0.001414 (0.001540) | 0.001695 (0.001772) | 0.002257 (0.002237) | 0.001341 (0.000832) | 0.001856 (0.000946) | 0.002887 (0.001175) |
| $\bar{y}^{*'}_{Pr}$ | 0.128960 (0.014889) | 0.156884 (0.018115) | 0.212731 (0.024569) | 0.139070 (0.016675) | 0.193736 (0.023231) | 0.303069 (0.036344) |
| $\bar{y}^{*'}_{BT}$ | 0.006911 (0.005275) | 0.008547 (0.006575) | 0.011818 (0.009175) | 0.008659 (0.007284) | 0.012382 (0.010476) | 0.019828 (0.016862) |
| $\bar{y}^{*'}_{SK}$ | 0.043198 (0.019510) | 0.051882 (0.023434) | 0.069251 (0.031282) | 0.040932 (0.019174) | 0.055692 (0.026072) | 0.085212 (0.039870) |
| $\bar{y}^{*'}_{D}$ | 0.001104 | 0.001356 | 0.001857 | 0.001259 | 0.001780 | 0.002811 |
| $\bar{y}^{*'}_{AH}$ | 0.009034 (0.019964) | 0.010769 (0.024295) | 0.014240 (0.032956) | 0.009898 (0.022392) | 0.010891 (0.031204) | 0.015979 (0.048828) |
| $\alpha = 0, \bar{y}^{*'}_{P1}$ | 0.001104 (0.000515) | 0.001356 (0.000633) | 0.001856 (0.000866) | 0.001259 (0.000609) | 0.001779 (0.000861) | 0.002809 (0.001360) |
| $\alpha = 1, \bar{y}^{*'}_{P1}$ | 0.001104 (0.000515) | 0.001356 (0.000633) | 0.001856 (0.000867) | 0.001259 (0.000609) | 0.001779 (0.000861) | 0.002809 (0.001360) |
| $\alpha = -1, \bar{y}^{*'}_{P1}$ | 0.001104 (0.000515) | 0.001356 (0.000633) | 0.001856 (0.000867) | 0.001259 (0.000609) | 0.001779 (0.000861) | 0.002809 (0.001360) |
| $\alpha_r = 1, r = 0, 1, 2, 3\ \bar{y}^{*'}_{GP}$ | **0.001104** (0.000515) | 0.001356 (0.000633) | 0.001856 (0.000867) | 0.000795 (0.000385) | 0.001712 (0.000829) | 0.002788 (0.001350) |
| $\alpha_0 = 0, \alpha_{1,2,3} = 1\ \bar{y}^{*'}_{GP}$ | 0.001104 (0.000515) | 0.001356 (0.000633) | 0.001857 (0.000867) | 0.000795 (0.000385) | 0.001712 (0.000829) | 0.002789 (0.001350) |

## 4.2 Application to real data set

In this section we have considered three data sets for numerical comparisons and results are given below. Population IV consists of 654 observations. The data summary is given below (see Tables 8).

**Population IV** [Source: [34]]

In Population IV, Forced expiratory volume is taken as the study variable, Age as the auxiliary variables and Smoke (No = 0, Yes = 1) is taken as scrambling response. The correlation coefficients are: $\rho_{ZX} = 0.7564$, $\rho_{XR_x} = 0.7831$ and $\rho_{ZR_x} = 0.6161$.

**4.2.1 Data collection.** To see the practical implication of measurement error, we conducted a study based on real data set at Quaid-i-Azam University, Islamabad during 2018. We distributed 55 questionnaires to the students of BS Statistics (5th Semester Fall, 2018) and M. Phil Statistics (1st and 2nd Semesters, Fall 2018). We consider our population of those students who gave the false response, which comes out to be 23. As we already have the true response from their academic record, which is available in the department of statistics. In question (i) we asked about $Y$ = Age and $X$ = Marks (in percentage) of Intermediate or Matric. In question (ii) $S$ = Social media effects the academic result is asked, where $Y$ is the study variable, $X$ is the auxiliary variable and $S$ is the scrambling response variable. We have 23 students ($N$ = 23), including 8 male students and 15 female students who gave the false response.

**Population V**. [Source: Section 4.2]

The explanation of the data set is given in the introduction Section 4.2.

$Y$: Age of BS $5^{th}$ and Mphil Students of Statistics department, $X$: Marks in O level or Matric, $S$: Social media effects on the academic result

**Table 8. Data summary.**

| Variable | Mean | St.Dev | Min | Med | Max |
|---|---|---|---|---|---|
| Forced expiratory volume ($Y$) | 2.63 | 0.86 | 0.79 | 2.54 | 5.79 |
| Age ($X$) | 9.93 | 2.95 | 3.00 | 10.00 | 19.00 |
| Smoke ($S$) 0,1 | 0.10 | 0.30 | 0.00 | 0.00 | 1.00 |

**Table 9. Mean squared error and |Bias| (in brackets) values for different estimators for Population IV with measurement error.**

| Estimators | 10% non-response | | | 20% non-response | | |
|---|---|---|---|---|---|---|
| | g | | | g | | |
| | 2 | 4 | 8 | 2 | 4 | 8 |
| $\bar{y}^{*'}_{HH}$ | 0.016275 | 0.020274 | 0.028272 | 0.018071 | 0.025662 | 0.040844 |
| $\bar{y}^{*'}_{R}$ | 0.011409 (0.000279) | 0.014217 (0.000310) | 0.019833 (0.000374) | 0.012629 (0.000304) | 0.017876 (0.000385) | 0.028370 (0.000549) |
| $\bar{y}^{*'}_{Pr}$ | 0.033931 (0.002057) | 0.041849 (0.002524) | 0.057686 (0.003458) | 0.037729 (0.002293) | 0.053244 (0.003231) | 0.084273 (0.005107) |
| $\bar{y}^{*'}_{BT}$ | 0.012244 (0.015038) | 0.015306 (0.019626) | 0.021431 (0.028803) | 0.013573 (0.017018) | 0.019295 (0.025567) | 0.030737 (0.042664) |
| $\bar{y}^{*'}_{SK}$ | 0.019332 (0.002895) | 0.023677 (0.003457) | 0.032367 (0.004580) | 0.021402 (0.003206) | 0.029886 (0.004389) | 0.046853 (0.006754) |
| $\bar{y}^{*'}_{D}$ | 0.011318 | 0.014124 | 0.019733 | 0.012531 | 0.017763 | 0.028224 |
| $\bar{y}^{*'}_{AH}$ | 0.012795 (0.002724) | 0.015843 (0.003343) | 0.021939 (0.004582) | 0.014156 (0.003036) | 0.019928 (0.004280) | 0.031472 (0.006767) |
| $\alpha = 0, \bar{y}^{*'}_{P1}$ | 0.011301 (0.004130) | 0.014097 (0.005152) | 0.019681 (0.007193) | 0.012510 (0.004572) | 0.017721 (0.006476) | 0.028118 (0.010276) |
| $\alpha = 1, \bar{y}^{*'}_{P1}$ | 0.011301 (0.004130) | 0.014097 (0.005152) | 0.019681 (0.007193) | 0.012510 (0.004572) | 0.017721 (0.006476) | 0.028118 (0.010276) |
| $\alpha = -1, \bar{y}^{*'}_{P1}$ | 0.011301 (0.004130) | 0.014097 (0.005152) | 0.019681 (0.007193) | 0.012510 (0.004572) | 0.017721 (0.006476) | 0.028118 (0.010276) |
| $\alpha_r = 1, r = 0, 1, 2, 3\ \bar{y}^{*'}_{GP}$ | **0.011258** (0.007126) | 0.014043 (0.008784) | 0.019600 (0.012122) | 0.012462 (0.007823) | 0.017647 (0.010935) | 0.027979 (0.017254) |
| $\alpha_0 = 0, \alpha_{1,2,3} = 1\ \bar{y}^{*'}_{GP}$ | 0.011116 (0.001680) | 0.013968 (0.000117) | 0.019619 (0.002986) | 0.012349 (0.000978) | 0.017637 (0.001945) | 0.028102 (0.007744) |

$N = 23, \bar{Z} = 23.78261, \bar{X} = 68.7391, \bar{R}_x = 12, S^2_Z = 3.996047, S^2_X = 73.65613,$
$S^2_Q = 0.723320, S^2_V = 41.25692, S^2_T = 0.2418484, \rho_{ZX} = 0.046204, \rho_{ZR_x} = -0.697343,$
$\rho_{XR_x} = 0.1429042.$

**Population VI**. [Source: Section 4.2]

The explanation of the data set is given in the introduction Section 4.2.

Y: Age of BS 5$^{th}$ and Mphil Students of Statistics department, X: Marks in A level or Intermediate, S: Social media effects on the academic result, Z = Y + S

$N = 23, \bar{Z} = 23.78261, \bar{X} = 60.3913, \bar{R}_x = 12, S^2_Z = 3.996047, S^2_X = 93.33992,$
$S^2_Q = 0.723320, S^2_V = 41.25692, S^2_T = 0.2418484, \rho_{ZX} = 0.117576, \rho_{ZR_x} = -0.697343,$
$\rho_{XR_x} = 0.026360.$

**Table 10. Mean squared error and |Bias| (in brackets) values for different estimators for Population IV without measurement error.**

| Estimators | 10% non-response | | | 20% non-response | | |
|---|---|---|---|---|---|---|
| | g | | | g | | |
| | 2 | 4 | 8 | 2 | 4 | 8 |
| $\bar{y}^{*'}_{HH}$ | 0.008117 | 0.010097 | 0.014058 | 0.009128 | 0.013128 | 0.021130 |
| $\bar{y}^{*'}_{R}$ | 0.003131 (0.000235) | 0.003882 (0.000253) | 0.005386 (0.000289) | 0.003540 (0.000251) | 0.005111 (0.0003014) | 0.008253 (0.000401) |
| $\bar{y}^{*'}_{Pr}$ | 0.025652 (0.002057) | 0.031514 (0.002524) | 0.043239 (0.003458) | 0.028640 (0.002293) | 0.040479 (0.003231) | 0.064156 (0.005107) |
| $\bar{y}^{*'}_{BT}$ | 0.004056 (0.016667) | 0.005090 (0.021759) | 0.007158 (0.031942) | 0.004593 (0.018977) | 0.006703 (0.028689) | 0.010923 (0.048112) |
| $\bar{y}^{*'}_{SK}$ | 0.010692 (0.002763) | 0.012869 (0.003284) | 0.017223 (0.004326) | 0.011879 (0.003047) | 0.016428 (0.004135) | 0.025527 (0.006313) |
| $\bar{y}^{*'}_{D}$ | 0.003065 | 0.003819 | 0.005325 | 0.003472 | 0.005041 | 0.008173 |
| $\bar{y}^{*'}_{AH}$ | 0.004413 (0.002725) | 0.005373 (0.003345) | 0.007294 (0.004585) | 0.004944 (0.003038) | 0.006967 (0.004282) | 0.011012 (0.006772) |
| $\alpha = 0, \bar{y}^{*'}_{P1}$ | 0.003063 (0.001119) | 0.003817 (0.001395) | 0.005321 (0.001944) | 0.003471 (0.001268) | 0.005037 (0.001841) | 0.008164 (0.002983) |
| $\alpha = 1, \bar{y}^{*'}_{P1}$ | 0.003063 (0.001119) | 0.003817 (0.001395) | 0.005321 (0.001944) | 0.0034713 (0.001268) | 0.005038 (0.001841) | 0.008164 (0.002983) |
| $\alpha = -1, \bar{y}^{*'}_{P1}$ | 0.003063 (0.001119) | 0.003817 (0.001395) | 0.005321 (0.001944) | 0.003471 (0.001268) | 0.005038 (0.001841) | 0.008164 (0.002983) |
| $\alpha_r = 1, r = 0, 1, 2, 3\ \bar{y}^{*'}_{GP}$ | **0.003042** (0.003096) | 0.003792 (0.003724) | 0.005288 (0.005012) | 0.003448 (0.003358) | 0.005007 (0.004584) | 0.008113 (0.007165) |
| $\alpha_0 = 0, \alpha_{1,2,3} = 1\ \bar{y}^{*'}_{GP}$ | 0.002826 (0.005569) | 0.003635 (0.004951) | 0.005210 (0.003709) | 0.003253 (0.005279) | 0.004896 (0.004094) | 0.008105 (0.001689) |

**Table 11. Mean squared error and |*Bias*| (in brackets) values for different estimators for Population V with measurement error.**

| Estimators | 10% non-response | | | 20% non-response | | |
|---|---|---|---|---|---|---|
| | g | | | g | | |
| | 2 | 4 | 8 | 2 | 4 | 8 |
| $\bar{y}_{HH}^{*'}$ | 0.867676 | 1.125663 | 1.64163 | 0.951954 | 1.378496 | 2.231579 |
| $\bar{y}_{R}^{*'}$ | 3.128806 (0.100601) | 4.254646 (0.134444) | 6.296395 (0.197716) | 3.758138 (0.118395) | 5.478096 (0.173855) | 8.918013 (0.284775) |
| $\bar{y}_{Pr}^{*'}$ | 3.654550 (0.005526) | 4.528472 (0.002878) | 6.486247 (0.001995) | 3.796421 (0.000402) | 5.618631 (0.001477) | 9.263050 (0.003626) |
| $\bar{y}_{BT}^{*'}$ | 1.367241 (174.9916) | 1.873681 (236.5221) | 2.781594 (349.1559) | 1.648715 (209.5474) | 2.385829 (307.1819) | 3.860058 (502.4508) |
| $\bar{y}_{SK}^{*'}$ | 10.43794 (0.307330) | 13.91542 (0.406211) | 20.45053 (0.595146) | 12.21497 (0.355589) | 17.91743 (0.523043) | 29.32235 (0.857952) |
| $\bar{y}_{D}^{*'}$ | 0.860832 | 1.124228 | 1.641161 | 0.951922 | 1.378200 | 2.230495 |
| $\bar{y}_{AH}^{*'}$ | 6.566224 (0.000407) | 8.815350 (0.006118) | 12.987039 (0.012305) | 7.758768 (0.0085405) | 11.35924 (0.011230) | 18.56018 (0.016611) |
| $\alpha = 0, \bar{y}_{P1}^{*'}$ | 0.859449 (0.036137) | 1.121871 (0.047171) | 1.636144 (0.068795) | 0.950229 (0.039954) | 1.374651 (0.057800) | 2.221205 (0.093396) |
| $\alpha = 1, \bar{y}_{P1}^{*'}$ | 0.859523 (0.036140) | 1.121997 (0.047177) | 1.636410 (0.068807) | 0.950322 (0.039958) | 1.374848 (0.057808) | 2.221725 (0.093418) |
| $\alpha = -1, \bar{y}_{P1}^{*'}$ | 0.859524 (0.036140) | 1.121998 (0.047177) | 1.636413 (0.068807) | 0.950322 (0.039958) | 1.374850 (0.057809) | 2.221733 (0.093418) |
| $\alpha_r = 1, r = 0, 1, 2, 3\ \bar{y}_{GP}^{*'}$ | **0.442375 (0.168156)** | 0.578775 (0.220813) | 0.811570 (0.310102) | 0.499340 (0.190152) | 0.698665 (0.265850) | 1.040157 (0.395165) |
| $\alpha_0 = 0, \alpha_{1,2,3} = 1\ \bar{y}_{GP}^{*'}$ | 0.788354 (0.031030) | 0.985573 (0.038702) | 1.343016 (0.052493) | 0.866074 (0.034045) | 1.173966 (0.045867) | 1.742806 (0.067544) |

Tables 9–14 show that the generalized class of proposed estimators $\bar{y}_{GP}^{*'}$ performs better than all other existing estimators both with and without measurement errors. The values of the absolute biases are given in brackets in Tables 9–14.

Table 9 shows that the generalized proposed estimator performs better than other estimators. The MSE for the generalized proposed estimator, when $\alpha_r = 1$, $r = 0, 1, 2, 3$ is 0.011258 for 10% non-response rate. When the non-response rate increases to 20%, the MSE for generalized proposed estimator increases to 0.012462. It is also observed that $\bar{y}_{R}^{*'}$ is less biased and $\bar{y}_{BT}^{*'}$ is most biased among all considered estimators. Table 10 shows the same pattern of results.

Table 11 shows that the generalized proposed estimator performs better than other estimators. The MSE for the generalized proposed estimator, when $\alpha_r = 1$, $r = 0, 1, 2, 3$ is 0.442375 for

**Table 12. Mean squared error and |*Bias*| (in brackets) values for different estimators for Population V without measurement error.**

| Estimators | 10% non-response | | | 20% non-response | | |
|---|---|---|---|---|---|---|
| | g | | | g | | |
| | 2 | 4 | 8 | 2 | 4 | 8 |
| $\bar{y}_{HH}^{*'}$ | 0.732836 | 0.947573 | 1.377047 | 0.803928 | 1.160847 | 1.874686 |
| $\bar{y}_{R}^{*'}$ | 2.085894 (0.062419) | 2.898353 (0.084903) | 4.313338 (0.125459) | 2.600062 (0.075925) | 3.776308 (0.111450) | 6.128800 (0.182501) |
| $\bar{y}_{Pr}^{*'}$ | 2.611639 (0.005526) | 3.172178 (0.002878) | 4.503190 (0.001995) | 2.638344 (0.000402) | 3.916842 (0.001477) | 6.473837 (0.003626) |
| $\bar{y}_{BT}^{*'}$ | 1.005383 (107.3366) | 1.401040 (148.7411) | 2.087388 (221.1230) | 1.248176 (134.2946) | 1.797146 (196.6074) | 2.895085 (321.2331) |
| $\bar{y}_{SK}^{*'}$ | 6.670812 (0.192784) | 9.024517 (0.257590) | 13.31206 (0.378374) | 8.026745 (0.228178) | 11.76322 (0.335830) | 19.23617 (0.551132) |
| $\bar{y}_{D}^{*'}$ | 0.722146 | 0.945328 | 1.376304 | 0.803877 | 1.160388 | 1.873005 |
| $\bar{y}_{AH}^{*'}$ | 4.234075 (0.003354) | 5.786298 (0.002295) | 8.564180 (0.006730) | 5.166670 (0.005263) | 7.550340 (0.006415) | 12.31768 (0.008719) |
| $\alpha = 0, \bar{y}_{P1}^{*'}$ | 0.721185 (0.030324) | 0.943683 (0.039679) | 1.372819 (0.057723) | 0.802686 (0.033750) | 1.157905 (0.048687) | 1.866539 (0.078483) |
| $\alpha = 1, \bar{y}_{P1}^{*'}$ | 0.721225 (0.030325) | 0.943751 (0.039682) | 1.372961 (0.057729) | 0.802736 (0.033753) | 1.158011 (0.048691) | 1.866820 (0.078495) |
| $\alpha = -1, \bar{y}_{P1}^{*'}$ | 0.721225 (0.030325) | 0.943751 (0.039682) | 1.372963 (0.057729) | 0.802736 (0.033753) | 1.158012 (0.048691) | 1.866824 (0.078495) |
| $\alpha_r = 1, r = 0, 1, 2, 3\ \bar{y}_{GP}^{*'}$ | **0.315665 (0.120889)** | 0.428155 (0.164937) | 0.611153 (0.235897) | 0.372965 (0.143547) | 0.527006 (0.202661) | 0.797297 (0.306103) |
| $\alpha_0 = 0, \alpha_{1,2,3} = 1\ \bar{y}_{GP}^{*'}$ | 0.646330 (0.025820) | 0.803842 (0.032048) | 1.074264 (0.042630) | 0.717330 (0.028638) | 0.954195 (0.037870) | 1.381019 (0.054364) |

**Table 13. Mean squared error and |Bias| (in brackets) values for different estimators for Population VI with measurement error.**

| Estimators | 10% non-response | | | 20% non-response | | |
|---|---|---|---|---|---|---|
| | g | | | g | | |
| | 2 | 4 | 8 | 2 | 4 | 8 |
| $\bar{y}_{HH}^{*'}$ | 0.867676 | 1.125663 | 1.641635 | 0.951954 | 1.378496 | 2.231579 |
| $\bar{y}_{R}^{*'}$ | 5.234362 (0.184108) | 6.785524 (0.238591) | 9.887849 (0.347557) | 5.046723 (0.179558) | 7.633174 (0.274687) | 12.80607 (0.464945) |
| $\bar{y}_{Pr}^{*'}$ | 5.281913 (0.000499) | 6.843380 (0.000608) | 9.966316 (0.000824) | 5.749089 (0.007383) | 8.745575 (0.011693) | 14.73854 (0.020313) |
| $\bar{y}_{BT}^{*'}$ | 1.953404 (251.5707) | 2.533396 (326.0361) | 3.693380 (474.9669) | 1.887851 (242.2096) | 2.803116 (370.3495) | 4.633645 (626.6292) |
| $\bar{y}_{SK}^{*'}$ | 18.38197 (0.552824) | 23.82296 (0.716382) | 34.70496 (1.043498) | 18.03340 (0.546057) | 27.50961 (0.835755) | 46.46204 (1.415150) |
| $\bar{y}_{D}^{*'}$ | 0.867644 | 1.125626 | 1.641589 | 0.945019 | 1.367140 | 2.211355 |
| $\bar{y}_{AH}^{*'}$ | 11.45452 (0.013466) | 14.84613 (0.017510) | 21.62934 (0.025598) | 11.16378 (0.002939) | 16.99394 (0.003907) | 28.65427 (0.005843) |
| $\alpha = 0, \bar{y}_{P1}^{*'}$ | 0.866182 (0.036420) | 1.123167 (0.047226) | 1.636364 (0.068805) | 0.943296 (0.039663) | 1.363519 (0.057332) | 2.201854 (0.092582) |
| $\alpha = 1, \bar{y}_{P1}^{*'}$ | 0.866314 (0.036426) | 1.123388 (0.047235) | 1.636833 (0.068824) | 0.943442 (0.039669) | 1.363841 (0.057346) | 2.202729 (0.092619) |
| $\alpha = -1, \bar{y}_{P1}^{*'}$ | 0.866315 (0.036426) | 1.123390 (0.047235) | 1.636838 (0.068825) | 0.943443 (0.039669) | 1.363844 (0.0573462) | 2.202743 (0.092619) |
| $\alpha_r = 1, r = 0, 1, 2, 3\ \bar{y}_{GP}^{*'}$ | **0.449115 (0.021302)** | 0.569335 (0.028025) | 0.783059 (0.041387) | 0.478928 (0.023557) | 0.661300 (0.034635) | 0.962034 (0.056514) |
| $\alpha_0 = 0, \alpha_{1,2,3} = 1\ \bar{y}_{GP}^{*'}$ | 0.798223 (0.030126) | 0.980530 (0.043517) | 1.331509 (0.073952) | 0.859641 (0.037758) | 1.166121 (0.055357) | 1.733732 (0.084949) |

10% non-response rate. When the non-response rate becomes 20%, the MSE for generalized proposed estimator increases to 0.499340. It is also observed that $\bar{y}_{AH}^{*'}$ is less biased and $\bar{y}_{BT}^{*'}$ is most biased among all considered estimators. Table 12 shows the same pattern of results.

Table 13 shows that the generalized proposed estimator performs better than other estimators. The MSE for the generalized proposed estimator, when $\alpha_r = 1$, $r = 0, 1, 2, 3$ is 0.449115 for 10% non-response rate. When the non-response rate becomes 20%, the MSE for generalized proposed estimator increases to 0.478928. It is also observed that $\bar{y}_{Pr}^{*'}$ is less biased and $\bar{y}_{BT}^{*'}$ is most biased among all considered estimators. Table 14 shows the same pattern of results.

Through numerical study it is concluded that the generalized proposed estimator performs better as compared to all other existing estimators. For 10% non-response rate the MSE value is minimum. The MSE values also increase as the value of constant g increases.

**Table 14. Mean squared error and |Bias| (in brackets) values for different estimators for Population VI without measurement error.**

| Estimators | 10% non-response | | | 20% non-response | | |
|---|---|---|---|---|---|---|
| | g | | | g | | |
| | 2 | 4 | 8 | 2 | 4 | 8 |
| $\bar{y}_{HH}^{*'}$ | 0.732836 | 0.947573 | 1.377047 | 0.803928 | 1.160847 | 1.874686 |
| $\bar{y}_{R}^{*'}$ | 3.016996 (0.103467) | 3.828422 (0.131635) | 5.451273 (0.187970) | 3.361892 (0.115596) | 4.863110 (0.168021) | 7.865545 (0.272871) |
| $\bar{y}_{Pr}^{*'}$ | 3.72329 (0.007424) | 4.827557 (0.010502) | 7.036093 (0.016659) | 4.126782 (0.008040) | 6.038035 (0.012350) | 9.860541 (0.020971) |
| $\bar{y}_{BT}^{*'}$ | 1.215590 (138.1246) | 1.542894 (175.2451) | 2.197501 (249.4861) | 1.347808 (154.4319) | 1.939547 (224.1670) | 3.123027 (363.6372) |
| $\bar{y}_{SK}^{*'}$ | 10.57577 (0.317827) | 13.47010 (0.405408) | 19.25877 (0.580571) | 11.80067 (0.354830) | 17.14482 (0.516415) | 27.83311 (0.839587) |
| $\bar{y}_{D}^{*'}$ | 0.721017 | 0.929116 | 1.344791 | 0.791492 | 1.140734 | 1.839091 |
| $\bar{y}_{AH}^{*'}$ | 6.565127 (0.002992) | 8.350240 (0.005370) | 11.92046 (0.010125) | 7.324078 (0.002967) | 10.62709 (0.005294) | 17.23311 (0.009948) |
| $\alpha = 0, \bar{y}_{P1}^{*'}$ | 0.720030 (0.030275) | 0.927484 (0.038998) | 1.341374 (0.056401) | 0.790305 (0.033230) | 1.138269 (0.047861) | 1.832686 (-0.077055) |
| $\alpha = 1, \bar{y}_{P1}^{*'}$ | 0.720096 (0.030278) | 0.927592 (0.039002) | 1.341599 (0.056410) | 0.790385 (0.033233) | 1.138437 (0.0478684) | 1.833125 (0.077078) |
| $\alpha = -1, \bar{y}_{P1}^{*'}$ | 0.720098 (0.030279) | 0.927593 (0.039003) | 1.341601 (0.056411) | 0.790386 (0.033233) | 1.138438 (0.047868) | 1.83313 (0.077078) |
| $\alpha_r = 1, r = 0, 1, 2, 3\ \bar{y}_{GP}^{*'}$ | **0.364646 (0.015332)** | 0.478386 (0.020114) | 0.704181 (0.029609) | 0.402961 (0.016949) | 0.592738 (0.024923) | 0.965155 (0.040582) |
| $\alpha_0 = 0, \alpha_{1,2,3} = 1\ \bar{y}_{GP}^{*'}$ | 0.653301 (0.027469) | 0.794303 (0.033398) | 1.052948 (0.044273) | 0.706101 (0.029689) | 0.939650 (0.039509) | 1.361983 (0.057268) |

## 5 Conclusion

In this study, we proposed a generalized class of estimators for the finite population mean when the variable of interest is stigmatizing in nature, considering both measurement error and non-response under simple random sampling. Through simulation study (see Tables 2–7) and real data sets (see Tables 9–14) it is observed that the proposed class of estimators $\bar{y}_{GP}^{*\prime}$ performs better than all existing estimators considered here.

## Supporting information

**S1 File. Data used in the manuscript "fevdata.csv".**
(CSV)

## Acknowledgments

The authors are grateful to the anonymous referees for their valuable comments and feedback.

## Author Contributions

**Conceptualization:** Javid Shabbir.

**Data curation:** Erum Zahid.

**Formal analysis:** Erum Zahid.

**Funding acquisition:** Ronald Onyango.

**Investigation:** Ronald Onyango.

**Methodology:** Erum Zahid.

**Project administration:** Erum Zahid.

**Resources:** Ronald Onyango.

**Software:** Erum Zahid.

**Supervision:** Javid Shabbir.

**Validation:** Javid Shabbir, Sat Gupta, Sadia Saeed.

**Visualization:** Javid Shabbir, Sat Gupta, Sadia Saeed.

**Writing – original draft:** Erum Zahid.

**Writing – review & editing:** Javid Shabbir, Sat Gupta, Sadia Saeed.

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
