## [Decision Letter · Decision Letter 0]

12 Oct 2021

PONE-D-21-27653A generalized class of estimators for sensitive variable in the

presence of measurement error and non-responsePLOS ONE

Dear Dr. Zahid,

Thank you for submitting your manuscript to PLOS ONE. After careful consideration, we feel that it has merit but does not fully meet PLOS ONE’s publication criteria as it currently stands. Therefore, we invite you to submit a revised version of the manuscript that addresses the points raised during the review process.

We look forward to receiving your revised manuscript.

Kind regards,

Maria Alessandra Ragusa, PhD Professor

Academic Editor

PLOS ONE

Journal Requirements:

Additional Editor Comments:

The paper needs major revision. Please, do it and send the revised version.

Best regards.

Reviewers' comments:

Reviewer's Responses to Questions

**Comments to the Author**

1. Is the manuscript technically sound, and do the data support the conclusions?

Reviewer #1: Partly

2. Has the statistical analysis been performed appropriately and rigorously? 

Reviewer #1: Yes

3. Have the authors made all data underlying the findings in their manuscript fully available?

Reviewer #1: No

4. Is the manuscript presented in an intelligible fashion and written in standard English?

Reviewer #1: No

5. Review Comments to the Author

Reviewer #1: In this papers, the authors studied a general class of estimators is proposed for the finite population mean of a sensitive variable. The paper has some weak points. The authors used some techniques based on classic estimators, without cite news trends in this fields (e.g., fractal-wavelet models). Thus I suggest to add (at least) the references below.

1. From Blackman–Tukey pilot estimators to wavelet packet estimators: a modern perspective on an old spectrum estimation idea. Signal Processing, 82(10), 1425-1441, 2002.

2. Evaluation for convergence of wavelet-based estimators on fractional Brownian motion. IEEE International Symposium on Information Theory, 2000.

3. Fractional-Wavelet Analysis of Positive definite Distributions and Wavelets on D′(C). In: Engineering Mathematics II, S. Silvestrov, M. Rancic (Eds.), Springer, pp. 337-353, 2016.

4. The mean consistency of wavelet density estimators. Journal of Inequalities and Applications, 2015(1).

Moreover, the English needs several improvements. I suggest a native English speaker. In particular, Abstract and Introduction cannot be publish in the current version. Finally, formulas must be rewritten in a more concise style.

6. PLOS authors have the option to publish the peer review history of their article (what does this mean?). If published, this will include your full peer review and any attached files.

Reviewer #1: No

---

## [Author Response · Author response to Decision Letter 0]

26 Nov 2021

Reviewer #1: In this papers, the authors studied a general class of estimators is proposed for the finite population mean of a sensitive variable. The paper has some weak points. The authors used some techniques based on classic estimators, without cite news trends in this fields (e.g., fractal-wavelet models). Thus I suggest to add (at least) the references below.

Suggested references are not relevant to the present paper.

Moreover, the English needs several improvements. I suggest a native English speaker. In particular, Abstract and Introduction cannot be publish in the current version. Finally, formulas must be rewritten in a more concise style.

Done

---

## [Editor Report · Decision Letter 1]

6 Dec 2021

A generalized class of estimators for sensitive variable in the

presence of measurement error and non-response

PONE-D-21-27653R1

Dear Dr. Zahid,

We’re pleased to inform you that your manuscript has been judged scientifically suitable for publication and will be formally accepted for publication once it meets all outstanding technical requirements.

Kind regards,

Maria Alessandra Ragusa, PhD Professor

Academic Editor

PLOS ONE

Additional Editor Comments (optional):

To the corresponding Author,

the revised version is now ready for publication.

Best regards.
---

## [Editor Report · Acceptance letter]

6 Jan 2022

PONE-D-21-27653R1 

A generalized class of estimators for sensitive variable in the presence of measurement error and non-response 

Dear Dr. Zahid:

I'm pleased to inform you that your manuscript has been deemed suitable for publication in PLOS ONE. Congratulations! Your manuscript is now with our production department. 

Kind regards, 

on behalf of

Dr. Maria Alessandra Ragusa 

Academic Editor

PLOS ONE